# Multi-Omics Elucidation of Flavor Characteristics in Compound Fermented Beverages Based on Flavoromics and Metabolomics

**DOI:** 10.3390/foods14234119

**Published:** 2025-12-01

**Authors:** Xiaolong Li, Jun Ma, Yannan Chu, Hui Li, Yin Zhang, Abo Li, Yonghua Jia

**Affiliations:** 1Horticultural Research Institute, Ningxia Academy of Agriculture and Forestry, Yinchuan 750002, China; xiaolong85115@163.com (X.L.);; 2College of Horticulture and Engineering, Shandong Agricultural University, Taian 271002, China; 19995272146@163.com

**Keywords:** compound fermented beverage, flavoromics, metabolomics, active substances, flavor characteristics

## Abstract

To characterize the key odorants and elucidate the flavor profiles of compound fermented beverages after fermentation, single-compound fermented beverages (GW, AW) and a compound fermented beverage (CW) were prepared using Italian Riesling grapes and SirPrize apples as raw materials. The flavor and metabolite profiles were systematically analyzed by integrating flavoromics (comprehensive two-dimensional gas chromatography–time-of-flight mass spectrometry, GC × GC–TOF MS) and metabolomics (LC–MS/MS). The results demonstrated that CW exhibited the most favorable acid/reducing sugars (2.18), imparting a drier taste and superior stability. Compounds with relative odor activity values (rOAV) greater than 1—including 3-methyl-1-butyl acetate, ethyl hexanoate, ethyl butanoate, and ethyl octanoate—collectively contributed prominent fruity, floral, and sweet aromas to all three wine types. Ethyl decanoate provided an additional distinctive traditional fruity note specifically to AW, while 1-octen-3-ol contributed a mushroom-like aroma to both GW and CW. Moreover, 3-methylbutanal, 4-ethyl-2-methoxyphenol, and ethyl 3-methylbutanoate added additional significant aroma contributions to CW, imparting floral, clove-like, and fruity notes, respectively. Notably, ethyl hexanoate (fruity aroma) exhibited a remarkably high rOAV of 27.43 in CW, significantly surpassing its levels in the single-substrate fermentations. Lipid metabolism and the phenylpropanoid pathway were significantly activated in CW, facilitating the coordinated synthesis of esters and phenolic compounds. Sensory attribute network analysis further confirmed that CW possessed more pronounced “sweet”, “fruity”, and “floral” characteristics. Correlation analysis revealed significant relationships between volatile organic compounds (VOCs) and total soluble solids (TS), titratable acidity (TA), the TA/TS ratio, and metabolite levels, underscoring the close connections among physicochemical properties, precursor/intermediate metabolites, and flavor formation. Comprehensive analysis of non-volatile metabolites and flavor-associated VOCs revealed variety-specific characteristics and compounding effects, providing valuable insights for enhancing the quality of compound fermented beverages.

## 1. Introduction

With the growing consumer demand for high-quality beverages, the compound fermented beverage market is trending towards flavor diversification and functionalization [1]. Apple cider is refreshing and sweet, while grape wine is rich in phenolic compounds and offers complex flavor profiles. Combining these two fruits for mixed-fruit fermentation is not merely a simple blending of flavors; rather, it leverages their distinct chemical compositions and metabolic substrates. Through yeast biotransformation, this approach aims to create a novel beverage with synergistic flavors, a balanced taste, and unique health benefits [2,3]. This process is complex, as the resulting flavor and nutritional profile represent a nonlinear outcome of microbial metabolism interacting with the raw material matrix [4]. Therefore, systematically elucidating the principles of flavor formation and metabolite characteristics in apple–grape compound fermentation is crucial for the precise design of product flavors and optimization of brewing processes.

The composition of compound fermented beverages includes both non-volatile matrix components and volatile compounds. Non-volatile components primarily contribute significant nutritional value, whereas volatile compounds are the decisive factors in forming the flavor characteristics and typicity of wine [5]. During fermentation, non-volatile aroma precursors existing in glycosidically bound forms in the fruit are hydrolyzed by enzymes from *Saccharomyces cerevisiae*, releasing free volatile compounds. These primarily include certain thiols and terpenes, which contribute to wine’s varietal aroma and impart its unique flavor profile [6]. Overall, the aroma of compound fermented beverages is largely composed of fermentation-derived compounds, which constitute the majority of the fragrance. These aroma compounds include esters, alcohols, organic acids, aldehydes, ketones, sulfur compounds, and terpenes [7]. Flavor substances derived from amino acid metabolism mainly include higher alcohols and their related esters, along with small amounts of aldehydes [8]. Following alcoholic fermentation, malolactic fermentation reduces acidity while generating diacetyl and ester compounds, further enhancing the wine’s flavor complexity [9].

The volatile compounds that constitute the aromatic profile of fruit wines are highly complex, encompassing a broad range of chemical classes such as alcohols, aldehydes, ketones, lactones, esters, terpenes, norisoprenoids, methoxypyrazines, and thiols [10]. Various methods are employed to identify flavor compounds in wines and fruit wines, including GC–MS, GC-O, GC-TOF/MS, and GC-IMS. GC-IMS is a straightforward preliminary process characterized by rapid detection, high sensitivity, minimal injection volume, low cost, and the ability to operate at atmospheric pressure without a vacuum system [11]. In a typical study, GC-O-MS was used to analyze the aromas of Chardonnay, Viognier, and Botrytis-affected Semillon wines, aiming to identify key esters associated with their fruity aroma profiles [12]. Notably, there is a limited number of cases that directly compare specific aroma differences among fruit wines. Flavoromics, as a systematic research strategy, integrates comprehensive analysis of both volatile and non-volatile compounds to decipher the overall chemical fingerprint of food flavor [13]. Comprehensive two-dimensional gas chromatography coupled with time-of-flight mass spectrometry (GC × GC-TOF MS) offers the distinct advantage of achieving ultra-high-resolution separation of the extremely complex volatile components in compound fermented beverages [14]. Compared to traditional one-dimensional GC, this technology significantly increases peak capacity, effectively separates co-eluting trace components, and greatly improves the qualitative accuracy and detection sensitivity for key aroma-active compounds (such as isoamyl acetate, linalool, and esters). This is crucial for precisely elucidating the synergistic or transformation pathways of aroma precursors during mixed-fruit fermentation [13,14,15]. Simultaneously, metabolomics provides a macro-perspective: utilizing liquid chromatography–mass spectrometry (LC-MS), it enables automatic identification of small molecule metabolites and systematic analysis of the dynamic patterns of sugars, acids, amino acids, and phenolics, as well as revealing their key roles in taste, stability, and health benefits [16,17,18].

Although multiple omics methods have been applied in the study of fruit wine, this study is the first to systematically reveal the flavor synergy mechanism between the unique Italian Riesling grape and SirPrize apple in the compound fermentation process in northwest China. This study employs an integrated systems biology approach, combining flavoromics and metabolomics, to comprehensively analyze the overall profile of an apple–grape compound fermented beverage post-fermentation. The objectives are as follows: (1) to identify and analyze potential aroma-active compounds in a compound dry compound fermented beverage using modern flavoromics techniques; (2) to estimate the relationship between aroma compounds and sensory attributes through multivariate statistical analysis, understand the aroma characteristics of compound (apple/grape) wines from the northwest region of China, and further verify the chemical basis of their sensory properties; and (3) to screen key odorants through relative odor activity value (rOAV) analysis and further validate them via aroma recombination experiments. Additionally, the possible formation pathways and mechanisms for several representative key odorants are revealed. The findings provide a theoretical foundation for the targeted regulation of high-quality compound fermented beverage fermentation.

## 2. Materials and Methods

### 2.1. Materials

Samples of SirPrize apples (high-acid, white-fleshed) and Italian Riesling grapes were collected from the National Apple Integrated Test Base (38°38′ N, 106°09′ E) and the National Grape Integrated Test Base (38°23′ N, 106°47′ E), respectively, at the Horticulture Research Institute of Ningxia Academy of Agriculture and Forestry Sciences (Ningxia, China). The region has a temperate continental climate, characterized by an annual precipitation of 200–220 mm and an average temperature ranging from 4 °C to 19 °C. Fruits were randomly harvested from trees with comparable fruit weight, tree architecture, and growth conditions. All selected apples and grapes were free from visible external defects, such as decay, scab, or insect damage. Fruit quality parameters are presented in Table 1. The selection of these fruit varieties is not only based on their local availability but, more importantly, on their complementary chemical composition. The Italian Riesling grape was selected for its ability to contribute complex varietal aromas, mainly terpenes and isoprene [19]. Sir Prize apples, as a high-acidity variety, are selected to enhance the acidity structure and crispness of the final beverage, while providing unique ester precursors that contribute to the fermentation aroma [20].

### 2.2. Wine Quality Analysis

Alcohol content (% *v*/*v*) was determined using a digital densitometer (DMA 35, Anton Paar, Graz, Austria). Total sugar content (g/L) was measured using Fehling’s reagent titration according to GB/T 15038-2006. Total acidity (expressed as tartaric acid, g/L) and volatile acidity (expressed as acetic acid, mg/L) were quantified using an automatic potentiometric titrator (916 Ti-Touch, Metrohm, Herisau, Switzerland). The pH was recorded with a laboratory pH meter (FE28, Mettler-Toledo, Greifensee, Switzerland). Free and total sulfur dioxide (mg/L) were determined via iodometric titration following the GB/T 15038-2006 standard method [21]. All measurements were performed in triplicate. Repeated independent biological fermentation was performed in triplicate (individual batches).

### 2.3. Brewing Methodology

According to the established protocol [22], the compound fermented beverage was produced using Italian Riesling grapes and SirPrize apples, a high-acidity, white-fleshed cultivar. The specific procedure was as follows: grape juice was extracted via pneumatic pressing (Willmes TYPE AC 20, Willmes, Lorsch, Germany), while apples were washed, sliced, and pulped to obtain apple slurry. The two components were mixed at a 4:1 ratio (CW), with control groups consisting of pure grape juice (GW) and pure apple slurry (AW). Each mixture was treated with pectinase (Lallzyme EX, Lallemand, Blagnac, France) at 0.1 g/L, PVPP (Polyclar VT, Ashland Corporation, Wilmington, DE, USA) at 0.3 g/L, and potassium metabisulfite (0.5 g/L SO_2_ equivalent), then held at 18 °C for 1–2 h. Commercial wine yeast Saccharomyces cerevisiae EC-1118 (Lallemand, France) was rehydrated following the standard protocol and inoculated at 20 g/hL (approximately 2 × 10^6^ CFU/mL). Sealed fermentation was carried out in 12 L stainless steel tanks (304 stainless steel, industry standard) at 18 ± 0.5 °C. Fermentation was considered complete when the specific gravity remained stable for three consecutive days and the residual sugar content dropped below 4.0 g/L, with the total process lasting approximately 14–16 days. After fermentation, the young beverage was racked into 12 L stainless steel tanks (304 stainless steel, industry standard) and aged at 18 ± 0.5 °C for 3 months. Finally, the beverage underwent sequential diatomite filtration, bentonite clarification (0.5 g/L), and sterile filtration through a 0.45 μm membrane before aseptic bottling. All fermentations were conducted in three independent biological replicates (*n* = 3). To ensure a complete fermentation, the yeast nutrition was supplemented with Diammonium Phosphate (DAP) at a total dose of 0.2 g/L. The DAP was added in two split doses: the first half at yeast inoculation, and the second half after 48 h of fermentation.

### 2.4. Flavor Compound Extraction and Analysis

Volatile compounds were extracted and analyzed following established methodologies [23,24,25,26]. Ethanol (99.8%, Aladdin, Shanghai, China), sodium chloride (AR, Sinopharm, Beijing, China), deuterated 1-hexanol-d13 (98.5%, C/D/N Isotopes, Pointe-Claire, QC, Canada), n-alkane standard (1000 mg/L, SIGMA, St. Louis, MO, USA), and n-hexane (GR, Yonghua, Suzhou, China) were used. A DVB/CAR/PDMS SPME fiber (57329-U, Supelco, Bellefonte, PA, USA) was employed for extraction. Instrumentation included an ultra-pure water system (Direct-8, Milli-Q, Darmstadt, Germany), a gas chromatograph (8890 A, Agilent, Santa Clara, CA, USA), and a time-of-flight mass spectrometer (Pegasus BT 4D, LECO, St. Joseph, MI, USA).

An internal standard (1-hexanol-d13) was prepared in 50% ethanol (*v*/*v*) at 10 mg/L. The n-alkane solution (1000 mg/L) was diluted with n-hexane to 10 mg/L for retention index calibration. All standard stock solutions were stored at 4 °C.

For extraction, wine samples were diluted with saturated NaCl solution to an ethanol concentration of 10% (*v*/*v*). A 5 mL aliquot of the diluted sample, spiked with 10 µL of the internal standard solution, was transferred to a 20 mL headspace vial. The SPME fiber was preconditioned at 270 °C for 10 min before extracting the sample headspace at 50 °C for 20 min with agitation. The fiber was then desorbed in the GC inlet at 250 °C for 5 min in splitless mode. The n-alkane standard mixture was analyzed under identical conditions for retention index determination.

Chromatographic separation was performed using an LECO Pegasus BT 4D GC × GC-TOF MS system equipped with an Agilent 8890A GC (Agilent, Santa Clara, CA, USA), a dual-stage jet modulator, and a split/splitless inlet (Agilent, Santa Clara, CA, USA). A DB-Heavy Wax column (30 m × 250 µm × 0.5 µm; Agilent, Santa Clara, CA, USA) served as the first dimension, coupled with a Rxi-5Sil MS column (2 m × 150 µm × 0.15 µm; Restek, Bellefonte, PA, USA) as the second dimension. High-purity helium was used as the carrier gas at a constant flow rate of 1.0 mL/min. The primary oven temperature program was as follows: 40 °C (hold 3 min), ramped at 6 °C/min to 200 °C, then at 10 °C/min to 250 °C (hold 5 min). The secondary oven was maintained 5 °C above the primary oven temperature. The modulator temperature was set 15 °C above the secondary oven with a modulation period of 4.0 s. The injector temperature was 250 °C.

Mass spectrometric detection was conducted with the TOF MS operated in electron ionization (EI) mode at 70 eV. The ion source and transfer line temperatures were both set at 250 °C. The acquisition rate was 200 spectra/s, with a detector voltage of 1960 V and a mass scan range of *m*/*z* 35–550.

Raw data were processed using Chroma TOF software (v.4.50). Peak identification required a signal-to-noise ratio (S/N) ≥ 50 and a spectral similarity match ≥ 700 against the NIST library. Retention index (RI) tolerances between calculated and theoretical values were set to <20 for reliable compound confirmation. Data were normalized to the internal standard, and compounds with >50% missing values were excluded from subsequent analysis. VOC classification was based on the PubChem database, and VOC–sensory attribute networks were visualized using the igraph R package (version 4.4.1) based on the FlavorDB database. Following the 3-month aging period in stainless steel tanks, all wine samples (GW, AW, CW) were aseptically bottled in 750 mL amber glass bottles sealed with screw caps. The bottled samples were stored in the dark at 4 °C until analysis. All chemical analyses (flavoromics and metabolomics) were completed within two weeks post-bottling to ensure the integrity of the volatile and non-volatile profiles.

### 2.5. Metabolite Extraction and Analysis

Metabolite extraction and analysis were performed according to established methodologies [27,28,29]. Methanol (≥99.0%, Thermo, San Jose, CA, USA), 2-chloro-L-phenylalanine (internal standard, 98%, Aladdin, Shanghai, China), acetonitrile (≥99.9%, Thermo, San Jose, CA, USA), formic acid (LC-MS grade, TCI, Tokyo, Japan), and ammonium formate (≥99.9%, Sigma) were used. Key instruments included a refrigerated centrifuge (H1850-R, Xiangyi, Hunan, China), a vortex mixer (BE-2600, Qilinbeier, TCI, Tokyo), a vacuum concentrator (5305, Eppendorf, Hamburg, Germany), and 0.22 μm PTFE filters (Jinteng, Tianjin, China).

For metabolite extraction, thawed samples were vortexed for 1 min. An aliquot was precisely transferred to a 2 mL tube, mixed with 400 μL of methanol, vortexed for 1 min, and centrifuged at 12,000 rpm (4 °C) for 10 min. The supernatant was completely transferred to a new tube and dried under vacuum. The residue was reconstituted in 150 μL of 80% methanol aqueous solution containing 4 ppm 2-chloro-L-phenylalanine. After filtration through a 0.22 μm membrane, the extract was subjected to LC-MS analysis.

Chromatographic separation was conducted on a Thermo Vanquish UPLC system using an ACQUITY UPLC^®^ HSS T3 column (2.1 × 100 mm, 1.8 μm; Waters, Milford, MA, USA) at 40 °C with a 0.3 mL/min flow rate and 2 μL injection volume. For positive ion mode, mobile phases consisted of 0.1% formic acid in acetonitrile (B2) and 0.1% formic acid in water (A2) with the following gradient: 0–1 min, 10% B2; 1–5 min, 10–98% B2; 5–6.5 min, 98% B2; 6.5–6.6 min, 98–10% B2; 6.6–8 min, 10% B2. For negative ion mode, acetonitrile (B3) and 5 mM ammonium formate in water (A3) were used with an identical gradient profile.

Mass spectrometric detection was performed using a Thermo Q Exactive mass spectrometer with electrospray ionization (ESI) in both positive and negative modes. Spray voltages were set at 3.50 kV (positive) and −2.50 kV (negative). The sheath and auxiliary gas flows were 40 and 10 arbitrary units, respectively. The capillary temperature was maintained at 325 °C. Full MS scans (*m*/*z* 100–1000) were acquired at 70,000 resolution, followed by HCD fragmentation at 30 eV for the top 10 ions at 17,500 resolution. Dynamic exclusion was applied to avoid redundant MS/MS acquisition.

Metabolite identification was achieved by matching MS and MS/MS data against public databases (HMDB, Massbank, KEGG, LipidMaps, LIPID MAPS Consortium San Diego, CA, USA, Mzcloud HighChem LLC Bratislava, Slovakia) and an in-house database (Panomix Biomedical Tech Co., Ltd., Suzhou, China). Molecular formulas were predicted from accurate mass and adduct information, followed by database matching. MS/MS spectral patterns were compared with database records for structural confirmation.

*p*-value correction method: For all *p*-values involved in multiple comparison hypothesis testing, we uniformly used the error detection rate control method; specifically, the Benjamini–Hochberg (BH) program. The metabolites ranked high in the importance index of variable projection obtained in the OPLS-DA model also passed the statistical significance verification of the BH-FDR correction mentioned above. The VIP value in the model is greater than 1.0, and the FDR value of intergroup differences is also less than 0.05. For metabolite extraction, aliquots of wine samples that had been stored at −80 °C were thawed overnight at 4 °C to ensure uniform thawing and minimize metabolite degradation.

### 2.6. Relative Odor Activity Value (rOAV) Analysis

The contribution of individual volatile compounds to the overall aroma profile was evaluated by calculating the relative odor activity value (rOAV) on a 0–100 scale, according to established methods [30]. The rOAV was defined as follows:rOAV = (OAV/OAVmax) × 100,
where OAVmax represents the highest odor activity value among all identified volatiles, and the OAV for a specific compound was calculated as OAV = C/OT, where C is the relative concentration of the compound and OT is its odor threshold in water. A compound was considered a key aroma contributor if rOAV > 1, and a modifier if 0.1 < rOAV ≤ 1. Generally, a higher rOAV indicates a greater contribution to the overall flavor.

### 2.7. Data Statistical Analysis

Flavor compounds: Data from the ChromaTOF software were integrated into a final compound table containing substance names, retention times, CAS numbers, database retention indices (RIs), experimentally determined RIs (C_7_–C_30_n-alkanes), and peak areas. Data were normalized to the total peak area or internal standard for cross-comparison.

Metabolites: Raw MS files were converted to mzXML format using ProteoWizard MSConvert (v3.0.8789). Peak detection, filtering, and alignment were performed with the R XCMS package (v3.12.0; parameters: bw = 2, ppm = 15, peakwidth = c(5, 30), mzwid = 0.015, mzdiff = 0.01, method = “centWave”). Normalization to the total peak area was applied to correct for systematic errors. Data visualization and analysis were conducted using custom R scripts, R base packages, Origin 2022, and TBtools (TBtools-II v 2.357).

PLS-DA was performed using SIMCA 13.0 (Umetricus, Umeå, Sweden) to classify and discriminate VOCs and non-VOCs. The developed models were cross-validated using default software options and interpreted by the variation (R2Y) and predictive ability (Q2). Metabolites with variable importance in projection (VIP) values greater than 1.0 and *p*-values less than 0.05 were selected as potential biomarkers for different types of fermented beverages. Metabolite concentrations were analyzed by one-way ANOVA (with fermented beverage type as the parameter); metabolites with *p*-values less than 0.05 were considered differential metabolites.

To correlate physicochemical indices with compounds, Pearson correlation coefficients (r) were calculated between the physicochemical indices and metabolites. Correlations with |*p*| > 0.6 and statistical significance (*p* < 0.05) were considered robust. The correlation networks and heatmaps were visualized using the classical Fruchterman-Reingold algorithm in Gephi (version 0.9.1, Web Atlas, Paris, France).

The integration of LC-MS-based metabolomics and GC × GC-TOF MS-based flavoromics data was performed at the level of biological interpretation, rather than direct data fusion. Each dataset was normalized and analyzed independently within its own platform to identify significant compounds. The resulting lists of significant non-volatile metabolites and volatile compounds were then integrated through correlation analysis and joint pathway mapping, in order to elucidate precursor–product relationships and systemic metabolic interactions. Apply internal standard normalization uniformly to all samples in the GC × GC-TOF MS dataset. Normalize the total peak area of non targeted metabolomics datasets.

## 3. Results

### 3.1. Basic Quality Parameters of the Three Compound Fermented Beverages

As shown in Figure 1, significant differences were observed in titratable acidity (TA), TA/TS ratio, total SO_2_(TSO_2_), and free SO_2_(FSO_2_). After fermentation, the compound fermented beverage (CW) exhibited significantly higher titratable acidity (6.96 g/L), resulting in a TA/TS ratio (2.18) significantly exceeding those of the single-varietal grape (GW) and apple (AW) wines. This indicates a drier, fresher taste and more structured mouthfeel for CW. No significant differences were identified in alcohol content or pH among the three groups, reflecting a stable and controlled fermentation process. Notably, CW showed the lowest pH (3.25), consistent with its higher acidity, which may contribute to aroma stability. Additionally, CW contained the highest levels of total and free SO_2_, suggesting enhanced antioxidant and antimicrobial potential beneficial for preserving complex volatile components. Similar volatile acidity across all wines indicated pure fermentation without contamination, supporting the reliability of the experimental data.

### 3.2. Flavor Compound Composition and Proportions in the Three Compound Fermented Beverages

Volatile profiles were analyzed using GC × GC-TOF MS. Stable baselines in total ion chromatograms and high overlap in peak intensities/retention times demonstrated analytical reliability (Appendix A). A total of 636, 662, and 615 volatile compounds were identified in the compound wine (CW), apple wine (AW), and grape wine (GW), respectively (Figure 2A). Alcohols, esters, and other compounds constituted the predominant volatile classes across all wines (Figure 2B). Specific compound distribution revealed the following: ketones (26 in CW, 25 in GW, 23 in AW); hydrocarbons (49, 51, 57); heterocyclic compounds (20, 17, 17); aldehydes (10, 14, 8); esters (97, 83, 109); alcohols (62, 63, 76); carboxylic acids (29, 34, 24); and other compounds (343, 328, 348) (Figure 2C). Venn analysis identified 253 volatile compounds common to all three wine types (Figure 2D). Given the low aroma activity of alkanes and minimal acid content, the differential composition of esters, terpenes, aldehydes, and alcohols primarily accounted for the distinct volatile profiles among the wines.

### 3.3. Multivariate Statistical Analysis of Flavor Compounds

Principal component analysis (PCA) of nine samples revealed clear separation among CW, AW, and GW groups, demonstrating appropriate sample selection and distinct flavor profiles (Appendix A). Loading analysis indicated that PC1 (32.3% variance) was primarily driven by terpenes and lipids, including γ-terpinene and ethyl 2-pentenoate, while PC2 (19.6% variance) correlated with alcohols such as 2-nonanol and 2-undecanol. The PCA model reliability was confirmed with R^2^X > 0.5 (Appendix A). PLS-DA further demonstrated significant separation among groups (Appendix A). Model validation parameters (R^2^X, R^2^Y, Q^2^) for pairwise comparisons ranged between 0.587 and 0.636, 1.00, and 0.963 and 0.973, respectively, with the three-group model achieving 0.517, 0.99, and 0.969 (Appendix A), confirming model robustness for distinguishing sample types. These results validate the analytical reliability and data quality for subsequent analyses.

### 3.4. Differential Flavor Compound Analysis

Differential flavor compounds were screened using criteria of VIP ≥ 1 and adjusted *p*-value < 0.05 from OPLS-DA models. Significant differences were observed in CW vs. GW (70 up-/62 down-regulated), CW vs. AW (141 up-/71 down-regulated), and GW vs. AW (126 up-/85 down-regulated) comparisons (Figure 3A). Hierarchical clustering of the 555 differential compounds revealed distinct expression patterns clearly separating the three wine types (Figure 3B). Venn analysis showed 91 compounds shared among all three comparison groups, with 17 common across four comparative sets (Figure 3C).

### 3.5. Characterization of Non-Volatile Compounds

Non-targeted metabolomic analysis was performed using LC-MS/MS. Quality assessment of total ion chromatograms from nine samples (three wine types) showed stable baselines and overlapping peaks in both positive and negative ion modes, indicating high analytical reproducibility (Appendix A). Principal component analysis revealed distinct clustering of replicate samples within AW, GW, and CW groups, demonstrating valid sample selection and inter-group metabolic differences (Appendix A).

PLS-DA further confirmed clear separation between groups (Appendix A). Model validation parameters for pairwise comparisons were as follows: GW vs. AW (R^2^X = 0.636, R^2^Y = 1.000, Q^2^ = 0.973), CW vs. AW (R^2^X = 0.635, R^2^Y = 1.000, Q^2^ = 0.970), and CW vs. GW (R^2^X = 0.587, R^2^Y = 1.000, Q^2^ = 0.963). All permutation tests showed Q^2^ regression line intercepts on the positive y-axis (Appendix A), validating model robustness for discriminating sample types. Using thresholds of FC > 2, VIP > 1, and *p*-value < 0.05, 1750 differential metabolites were identified. Pairwise comparisons revealed 509 (214 up/295 down) in CW vs. GW, 583 (339 up/244 down) in CW vs. AW, and 658 (366 up/292 down) in GW vs. AW (Figure 4A).

KEGG enrichment analysis of differential metabolites highlighted phenylpropanoid biosynthesis, linoleic acid metabolism, and α-linolenic acid metabolism as key pathways (Figure 4B). These were integrated with terpenoid backbone and carotenoid biosynthesis pathways related to ester, phenolic, aldehyde, and alcohol formation. Subsequent analysis combined rOAV values with differential metabolite patterns to construct provisional metabolic profiles connecting precursor compounds and aroma-active metabolites. Hierarchical clustering confirmed distinct metabolic patterns among wine types (Appendix A). Volatile esters (fruity aroma): Compounds such as ethyl caproate, ethyl butyrate, and isoamyl acetate (VIP > 1.0) are significantly higher in apple containing fermented products (AW and CW). This strongly indicates that lipid-metabolism- and amino-acid-derived esterification pathways are enhanced in the apple matrix. Apple cider is rich in fatty acid precursors and specific amino acids, which seems to provide a more favorable environment for yeast to synthesize these key fruit esters. Advanced alcohols (complexity and fusel annotation): The characteristic controlled by wine (GW) is a higher relative abundance of isopentanol and 2-phenylethanol. This is consistent with the highly active Ehrlich pathway during grape juice fermentation, where the degradation of branched and aromatic amino acids (abundant in grapes) leads to the formation of these higher alcohols. Terpenes and norepinephrine (variety characteristics): As expected, several linalool oxides and β-Damascenone in the GW and CW groups have high loadings. This directly reflects the unique secondary metabolism of grapes, as these compounds originate from the terpenoid and carotenoid pathways in grapes.

### 3.6. Network Interactions Between VOCs and Sensory Aroma Attributes

Network analysis utilizing the FlavorDB database and igraph package visualized relationships between volatile organic compounds (VOCs) and sensory aroma attributes. Figure 5A and Appendix A demonstrate that significant VOCs contribute to the top 10 aroma attributes, including “sweet,” “fruity,” “green,” “waxy,” “ethereal,” and “floral.” The network reveals that individual aroma attributes are modulated by multiple compounds, while single VOCs often contribute to multiple aroma characteristics. Consequently, changes in VOC abundance may alter the overall aroma profile, though such effects could be neutralized or compensated by other compounds.

The aroma profiles of the three wine types are presented in Figure 5B. Notably, the compound fermented beverage (CW) exhibited more pronounced “sweet,” “fruity,” “apple,” and “floral” attributes, potentially explaining its perceived richer and more complex flavor profile.

### 3.7. Compound Fermented Beverage Aroma rOAV

Figure 6 shows the relative odor activity values (ROAV ≥ 0.1) of the 34 volatile aroma compounds identified in the compound fermented beverages. Although the three beverage samples shared similar volatile aroma compounds, their relative contributions varied substantially, resulting in distinct aroma profiles. The results revealed that eight key compounds (ROAV > 1) dominated the overall aroma profile (Figure 6A). Among them, esters played a central role: isoamyl acetate and ethyl butyrate exhibited high activity across all samples, while ethyl hexanoate (intense fruity aroma) had a significantly higher contribution (ROAV = 27.43) in the compound beverage (CW) than in single-substrate fermentations, indicating a synergistic effect arising from the mixed apple and grape fermentation. Additionally, 1-octen-3-ol and 4-ethyl-2-methoxyphenol, which was exclusively present in CW, together contributed to the unique aroma complexity of the compound beverage. In contrast, the GW sample was characterized by an exceptionally prominent banana-like fruit aroma imparted by isoamyl acetate. The CW sample exhibited the most complex aroma combination, characterized by high levels of ethyl hexanoate, 1-octen-3-ol, and 4-ethyl-2-methoxyphenol. Furthermore, esters were the major contributors to the aroma of all samples, but the relative dominance of different esters suggested potential differences underlying the fermentation processes. It is noteworthy that ethyl octanoate possessed the highest ROAV (ROAV = 40) in all samples, indicating it is the most critical aroma compound in these beverages.

Among the secondary contributors (0.1 < RO ≤ 1, Figure 6B), ethyl 2-methylpropanoate exhibited the highest activity in CW, further enhancing its fruity character. Meanwhile, acids such as octanoic acid and hexanoic acid contributed supportive soapy base notes, yet their relatively low ROAV values ensured the overall balance of the flavor profile.

In total, 34 volatile aroma compounds (rOAV > 0.1) (Appendix A) play a crucial role in determining the aroma and flavor of compound fermented beverage, which is a key attribute affecting consumer preferences and market value. These compounds are mainly synthesized through specific metabolic pathways of fatty acids, amino acids, and carbohydrates (Figure 7). The biosynthesis of straight-chain aldehydes, alcohols, and esters originates from lipid precursors (mainly linolenic acid/linoleic acid), which are broken down via beta oxidation and catalyzed by lipoxygenase (LOX). Among them, the CW type compound fermented beverage in the LOX pathway contains high levels of 2-methyl-propanoic acid ethyl ester (C6) and heptanoic acid ethyl ester (C9) lipid volatile compounds, while the butanoic acid ethyl ester and hexanoic acid propyl ester lipid volatile compounds are at high levels in GW, both higher than AW levels. At the same time, branched variants specifically appear through the isoleucine catabolic cascade, forming characteristic secondary metabolites, with AW-type compound fermented beverage acidic volatile compounds at low levels. The production of terpenoids is carried out through a dual enzyme system involving the metabolism framework of mevalonate and methylerythritol phosphate, while phenylpropanoids are generated through a phenylpropanoid network derived from shikimic acid. The resulting phenolic volatile compound 4-ethyl-2-methoxy-phenol is at a high level in GW and CW compound fermented beverages, and 4-ethyl-phenol shows a high level in AW compound fermented beverages.

The main flavor compounds with rOAV > 1 are the ones that contribute significantly to the overall flavor (Figure 8, Appendix A). The rOAVs of the main flavor compounds in AW, GW, and CW are as follows: 6 (3-methyl-1-butanol acetate, hexanoic acid ethyl ester, decanoic acid ethyl ester, butanoic acid ethyl ester, octanoic acid ethyl ester), 5 (1-Octen-3-ol, 3-methyl-1-butanol acetate, hexanoic acid ethyl ester, butanoic acid ethyl ester, octanoic acid ethyl ester), 8 (1-Octen-3-ol, 3-methyl-butanal, 4-ethyl-2-methoxy-phenol, 3-methyl-butanoic acid ethyl ester). Ethyl ester, 3-methyl-1-butanol acetate, hexanoic acid ethyl ester, butanoic acid ethyl ester, octanoic acid ethyl ester). The increase in the number of main flavor compounds in CW (compound fermented beverage) indicates that the compounds that contribute significantly to the overall flavor of CW type are concentrated in the combined fermentation of apples and grapes.

### 3.8. Correlations Among VOCs, Wine Physicochemical Properties, and Metabolites

Correlation analysis between physicochemical indices and 34 key aroma compounds revealed significant associations primarily with TS, TA, TA/TS, TSO_2_, and FSO_2_ (Figure 9). Most aroma compounds showed minimal correlation with ABV, pH, and VA. Exceptions included ethyl heptanoate and octanal, which exhibited significant negative correlations with ABV (*p* < 0.05), and 3-methylbutyl acetate, which correlated negatively with VA (*p* < 0.05) (Figure 9A, Group I). Significant correlations between physicochemical indices and both VOC content and rOAV values were mainly concentrated in Groups I and III. Compounds including 3-methylbutanal, ethyl lactate, 1-butanol, 4-ethyl-2-methoxyphenol, 4-ethylphenol, ethyl propanoate, diethyl succinate, and butanoic acid demonstrated strong correlations (Figure 9A,B). Although 2-heptanol, nonanoic acid, and hexanoic acid showed high correlation with physicochemical indices (Figure 9A), their contribution to flavor perception decreased due to higher odor thresholds (Figure 9B). Conversely, 1-octen-3-ol, benzyl alcohol, 1-propanol, octanal, ethyl 3-methylbutanoate, ethyl hexanoate, and ethyl 2-methylpropanoate exhibited low content–physicochemical correlation but high rOAV–physicochemical correlation, indicating threshold-dependent flavor contribution.

In order to better understand the formation of unique flavors in different types of compound fermented beverages, the relationship between LC-MS (non VOCs) metabolome and GC × GC-TOF MS was elucidated (Figure 8). Overall, volatile compounds, including alcohols, esters, aldehydes, acids, phenols, and some heterocyclic compounds, exhibit significant positive or negative correlations with various non-volatile metabolites. These correlations indicate that the formation of these VOCs depends on the presence of metabolic precursors. In this study, a large number of VOCs were accurately identified using GC × GC-TOF MS, with the majority of VOCs coming from the catabolism of upstream metabolites. Flavor precursors undergo various processes such as enolization, decarboxylation, and degradation to produce furan, aldehyde, ketone, thiol ketone aldehyde, and other compounds. p-Coumaraldehyde, coniferyl acetate, cinnamic acid, sinapaldehyde, FA 20_3; O4, sinapine, trans-Coumaryl acetate, isopentenyl pyrophosphate, and cis-3-Hexenyl acetate non-volatile compounds are significantly correlated with many VOCs, and most of them are positively correlated (Group I). By contrast, glutaric acid, coniferin, 13-L-hydroperoxylinoleic acid, L-tyrosine, syringin, caffeyl alcohol, chavicol, farnesyl pyrophosphate, methyl jasmonate, 11-HpODE, lipoxin B4, farnesol, chlorogenic acid, (Z)-3-hexenal, and geranyl-PP metabolites are significantly positively correlated with VOCs (Group II). These results indicate that these non-volatile metabolites may be important flavor precursors, and their content is a key factor in the development of complex compound fermented beverage flavors. Many of these flavor compounds and intermediates have their own unique flavors.

## 4. Discussion

The compound fermented beverage’s flavor quality is intrinsically linked to its market acceptance [31]. This study systematically elucidated the impact of compound fermentation on the basic physicochemical properties, volatile flavor profiles, and sensory characteristics of wines produced from grapes, apples, and a grape–apple blend. By integrating basic quality analysis, flavoromics, and non-targeted metabolomics, we assessed product quality differences and investigated the underlying metabolic pathways and material basis.

### 4.1. Differences in Basic Physicochemical Properties

The unique physicochemical properties of the compound fermented beverage (CW) formed the foundation for its flavor and stability. CW exhibited a significantly higher acid/reducing sugars than GW and AW (Figure 1), contributing directly to a drier, fresher taste and more structured mouthfeel [32]. This is consistent with observations in other fruit fermentation systems; for instance, in pomegranate-grape composite wine (PGCW), the physicochemical properties were highly correlated with its metabolite profile, where pH and titratable acidity were directly related to organic acid metabolism [33]. Similarly, in the development of a compound beverage from Actinidia arguta and Aronia melanocarpa, the careful balance of fruit pulp and citric acid addition was crucial to achieving a sour, sweet, and delicious taste with a unique fruity aroma [34]. Its lowest pH value not only intensified the perception of acidity but also inhibited microbial growth, enhancing the stability of flavor and color compounds [35]. Concurrently, CW’s significantly higher sulfur dioxide content provided greater antioxidant and antimicrobial potential, crucial for protecting complex aroma compounds—particularly easily oxidized terpenes and thiols—from deterioration [36]. The protection of aroma compounds is a key concern in beverage development. In the study of mulberry-Vitis amurensis compound beverage, electronic nose and GC-IMS analysis confirmed significant differences in flavor profiles between single and compound beverages, with 17 compounds being more abundant in the composite beverage, contributing to its pleasant flavor [37]. This highlights the importance of preserving such complex aroma profiles, which can be susceptible to oxidation, through the use of antioxidants like sulfur dioxide.

### 4.2. Metabolic Pathways of Aroma Compounds

Fatty acids are primary precursors for aroma volatiles in fruit. Both β-oxidation and the lipoxygenase (LOX) pathway are key enzymatic systems for fatty acid catabolism, generating C1–C20 straight-chain aldehydes, alcohols, ketones, acids, and esters that significantly impact fruit aroma [38]. This is consistent with findings in Baijiu fermentation, where extracellular proteolytic enzymes and β-oxidation were pivotal for generating raspberry and creamy flavors [39]. β-oxidation plays a fundamental role during fermentation, primarily occurring in peroxisomes in plants [40]. Fatty acids are activated to acyl-CoAs, which are sequentially shortened, requiring FAD, NAD, and free CoA. Acyl-CoAs can be reduced to aldehydes and then to alcohols through alcohol dehydrogenase (ADH), enabling alcohol acyltransferase (AAT) to produce esters [41]. Ester production depends on the supply of acyl-CoAs from β-oxidation and alcohols. AAT’s broad specificity for alcohols and acyl-CoAs explains the diversity of esters synthesized [42]. Key esters like ethyl hexanoate and ethyl octanoate were significantly more abundant in CW, providing direct evidence of their formation via β-oxidation followed by AAT catalysis. The enrichment of branched-chain esters like ethyl 2-methylpropanoate in CW suggests that compound fermentation alters the acyl-CoA pool, guiding AAT towards the synthesis of a more diverse ester profile.

The LOX pathway is also crucial for aroma formation (Figure 7). Linoleic (C18:2) and α-linolenic (C18:3) acids are primary LOX substrates [43]. LOX enzymes (9-LOX and 13-LOX) produce corresponding hydroperoxides, which are cleaved by hydroperoxide lyases (HPL) into C9 and C6 aldehydes (green leaf volatiles) [44]. These aldehydes are metabolized by ADH to alcohols and subsequently esterified by AAT [45]. Metabolomic data showed enrichment of linoleic and α-linolenic acid metabolism pathways in CW, providing the substrate basis for LOX activation. Significant enrichment of 1-octen-3-ol, derived from unsaturated fatty acid oxidation, indirectly demonstrates LOX pathway activity in CW [46], adding mushroom/earthy complexity. It is worth noting that lipid-derived molecular markers, such as lactones and ketones resulting from fatty acid oxidation, have been recognized as crucial odorants in studies on premature aging of red wines, underscoring the broad significance of lipid oxidation in shaping beverage aroma profiles [47]. KEGG enrichment revealed intertwined “terpenoid backbone biosynthesis” and “phenylpropanoid biosynthesis” pathways with fatty acid metabolism [48]. For instance, the synergistic increase in phenols like 4-ethyl-2-methoxyphenol (rOAV = 1.48) and esters explains CW’s unique flavor profile combining fruity and spicy notes. These findings clarify the advantage of compound fermentation at the metabolic level: the differential fatty acid precursors from apple and grape, rewired through yeast metabolism, generate flavor diversity exceeding that of single-compound fermented beverages, providing a theoretical basis for targeted flavor optimization [49]. A similar phenomenon was observed in a pomegranate-grape composite fruit wine (PGCW), where the unique metabolite profile and interactions between different metabolic pathways (e.g., amino acid and secondary metabolism) were also attributed to the composite fermentation process [34].

### 4.3. Evaluation of Aroma Compounds via rOAVs

Quantification based on relative odor activity values (rOAVs) clearly revealed quality differences. The synergistic effect of eight key flavor compounds (rOAV > 1) in CW formed the core structure of its aroma. This approach aligns with the methodology employed in the molecular characterization of kava flavor, where relative odor activity values were instrumental in identifying 22 key aroma-active compounds and elucidating its distinctive flavor profile [50]. The unique combination of 3-methylbutanal (malty), 4-ethyl-2-methoxyphenol (spicy), and esters (fruity) created a multi-layered volatile evolution (Figure 6A and Appendix A) [51], a hallmark of high-quality fermented beverages.

The distribution pattern of esters is notable. Medium-chain esters like ethyl hexanoate and ethyl octanoate maintained high rOAVs across all wines, confirming their foundational role. However, CW expanded the volatile spectrum further by supplementing with branched-chain esters like ethyl 3-methylbutanoate [52]. This “mainstream + specialty” ester combination allowed CW to retain the familiar fruity base while adding pleasant volatile variations.

AW’s rOAV profile exposed its limitations. Although ethyl decanoate’s exclusive high contribution provided a specific fruity note, the overall lower rOAVs for most compounds resulted in weaker volatile intensity (Figure 6B and Appendix A) [53]. In contrast, GW’s significant contribution from 1-octen-3-ol (mushroom) defined a specific regional style, but this single dominant compound somewhat limited volatile complexity [54].

The sensory descriptors “sweet” and “fruity” prominent in CW correlated directly with the balanced ratio of esters and terpenes [55]. The co-occurrence of “apple” and “floral” descriptors (Figure 5B) indicated that CW successfully fused the freshness of apple wine with the elegance of grape wine, demonstrating the direct value of the compounding process.

The superior flavor complexity of CW holds significant implications for industry innovation. The terpene glycosides and phenolics from grapes, combined with ester precursors and organic acids from apples, catalyzed by fermenting microbes, produced a synergistic flavor effect (1 + 1 > 2) [56,57]. This effect is manifested not only in the increased number of key compounds (8 in CW, Figure 6A), but also in the leap in volatile quality. The unique flavor compound profile in CW—such as the specific combination of branched-chain esters and terpenes, which is unavailable when using single fruits—offers a technical pathway for developing novel flavor products with IP potential [58].

### 4.4. Research Value of Compound Fermented Beverage

The flavor balance exhibited by CW holds considerable market value. The excessive strength of certain character compounds in AW and GW [30] was effectively mitigated in CW. This balance makes CW more suitable as a base wine for subsequent aging or flavor blending, showing greater product extensibility. In sparkling wine production, for instance, the quality and balance of the base wine are critical, as its characteristics can predict the development of potential flaws, like atypical aging, in the final product after secondary fermentation and aging [59].

CW’s combination of familiarity and novelty—retaining common esters like ethyl hexanoate and ethyl octanoate for a familiar fruity base [60], while incorporating novel compounds like 3-methylbutanal and 4-ethyl-2-methoxyphenol [49]—reduces consumer trial barriers while satisfying curiosity, offering clear marketing advantages. Future research should focus on optimizing fruit ratios and fermentation parameters to enhance the generation of desirable flavor compounds and explore different grape–apple combinations. The transformation of these compounds during storage and their quantitative relationship with sensory evaluation are also important topics.

### 4.5. Link Between Flavor and Matrix: Correlations of rOAV Compounds with Physicochemical Indices

The physical and chemical indicators of the compound fermented beverage not only define its basic characteristics but also constitute the macro environment for the formation and expression of flavor compounds. Through correlation analysis, the complex and important intrinsic relationship between key aroma compounds (rOAV > 0.1) and basic physicochemical indicators was revealed, and the physicochemical driving factors of the flavor advantage of the compound fermented beverage (CW) were elucidated [61]. The analysis results (Figure 9) clearly indicate that the aroma compounds that contribute the most to flavor (measured by their rOAV values) have a significant correlation network with a specific set of physicochemical indicators.

Total acid (TA), total sugar (TS), acid/reducing sugars (TS/TA), and sulfur dioxide (total and free) are the core nodes of this network. In particular, the high TA and TS/TA ratios that give CW a dry and fresh taste are significantly positively correlated with various key flavor compounds. For example, 3-methylbutanal, which contributes to malt aroma, and 4-ethyl-2-methoxyphenol, which contributes to a pungent aroma, are highly positively correlated with TA and TS/TA. This discovery holds dual significance: firstly, it explains the high consistency between taste and aroma in CW from a chemical perspective—a high-acidity environment not only provides a refreshing framework but also accompanies the enrichment of certain specific aroma substances [52,62]; secondly, it suggests that the fermentation metabolism of CW may tend to produce more organic acids and certain branched aldehydes and phenols simultaneously, which may be the common metabolic basis for its unique flavor formation [63]. CW has the highest sulfur dioxide content, which not only directly endows it with excellent antioxidant and antibacterial potential but also creates a stable chemical environment that is positively correlated with the preservation and abundance of a series of ester substances contributing to the “fruit” and “sweet” aromas. This indicates that the higher levels of sulfur dioxide in CW act as a “protective shield,” effectively protecting the volatile aroma compounds that are susceptible to oxidation and microbial invasion [64], allowing them to be better preserved after fermentation and even during aging, thereby directly supporting their more prominent fruity and floral contours (Figure 5). The critical importance of a stable fermentation environment and microbial activity for flavor development is further underscored by research on Semillon wine, which highlighted how functional microorganisms, including Saccharomyces and Hanseniaspora, are positively associated with most core flavor compounds [65]. It is particularly noteworthy that the correlation between alcohol content (ABV) and volatile acids (VAs) and the vast majority of key flavor compounds is weak. This confirms the early judgment of this study that the stimulating taste and acidity intensity are not driven by alcohol content, and also indicates that the differences in flavor among the three groups of compound fermented beverage are mainly not due to fermentation health status (because the VA content is low and there is no difference), but rather to more refined metabolic regulation, such as the production of furan, aldehyde, ketone, thiol ketone aldehyde, and other compounds through various processes such as enolization, decarboxylation, and degradation of flavor precursors (Figure 10).

In summary, correlation analysis has drawn a clear picture of “matrix-driven flavor”: the unique high acid, high acid/reducing sugars core taste architecture, and stable chemical environment of high sulfur dioxide in CW not only directly defines its dry and fresh taste tone, but also creates and maintains an internal microenvironment which is conducive to the formation and stability of key aroma compounds (especially aldehydes, phenols, and easily oxidizable esters that contribute complexity) [64,65,66,67]. This reveals, from the perspective of system correlation, that the excellent flavor quality of CW is not accidental but rather the inevitable result of the synergy and joint action of its inherent physicochemical properties and metabolites.

Based on the ROAV method, this study successfully identified volatile compounds with potential key contributions to the overall aroma of the compound fermented beverage. However, it should be noted that a key limitation lies in the fact that these inferences are primarily derived from theoretical calculations and literature-based odor thresholds. Although ROAV analysis serves as a robust and widely recognized predictive tool, final confirmation of aroma perception must be achieved through direct sensory evaluation, such as aroma recombination and omission experiments. Therefore, we recommend that future studies focus on sensory validation to definitively confirm the actual impact of these compounds on the overall flavor profile.

## 5. Conclusions

This study systematically elucidated the physicochemical and metabolic basis for the superior flavor quality of a compound fermented beverage (CW) compared to traditional single-compound fermented beverages. The characteristically higher acidity-to-reducing sugars of CW established its dry and refreshing taste profile, while elevated sulfur dioxide levels contributed to flavor stability. Flavoromics analysis identified a greater diversity of key aroma compounds (rOAV > 1) in CW, with more prominent fruity and floral sensory attributes, demonstrating a synergistic flavor effect from mixed-fruit fermentation. Metabolomics revealed that this enhancement was attributed to the activation of metabolic pathways including lipid and phenylpropanoid biosynthesis, facilitating coordinated synthesis of esters and phenolics. Correlation analysis further confirmed positive relationships between the high-acidity environment and the formation of key aldehydes and phenols, establishing intrinsic connections between flavor compounds and the wine matrix. In conclusion, compound fermentation reshapes the metabolic network and physicochemical microenvironment, resulting in a novel compound fermented beverage product with enhanced flavor complexity and stability, demonstrating significant research value and application potential.

## Figures and Tables

**Figure 1 foods-14-04119-f001:**
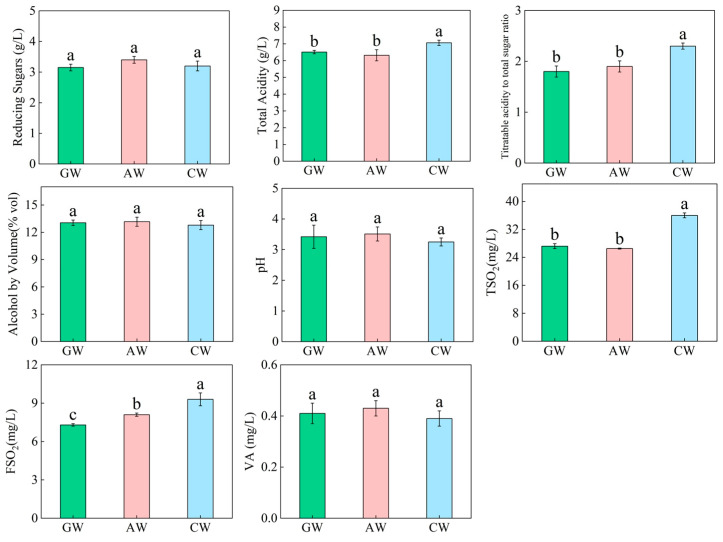
Physicochemical parameters of the three alcoholic beverages. Different lowercase letters indicate significant differences (*p* < 0.05). TS: reducing sugars; TA: titratable acidity; TA/TS: titratable acidity to reducing sugars; ABV: alcohol by volume; TSO_2_: total sulfur dioxide; FSO_2_: free sulfur dioxide; VA: volatile acidity. GW: grape wine; AW: apple cider; CW: compound fermented beverage. Significant differences were observed through one-way ANOVA and post hoc Tukey HSD test (*p* < 0.05). *n* = 3.

**Figure 2 foods-14-04119-f002:**
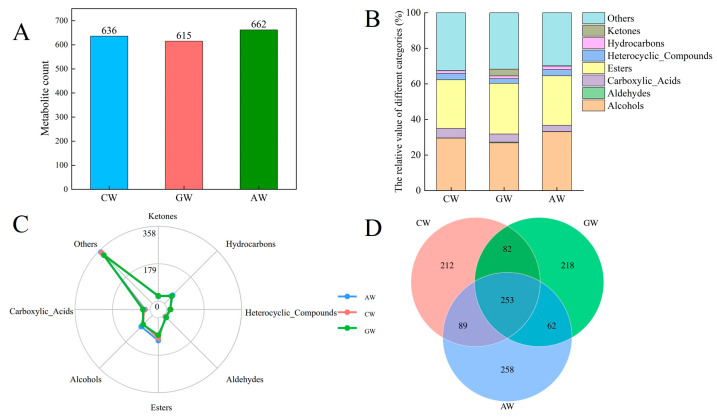
Flavor compounds in the three alcoholic beverages. (**A**) Number of volatile compounds across samples. (**B**) Relative content of compound classes. (**C**) Radar chart showing volatile distribution by chemical class. (**D**) Venn diagram of shared and unique compounds.

**Figure 3 foods-14-04119-f003:**
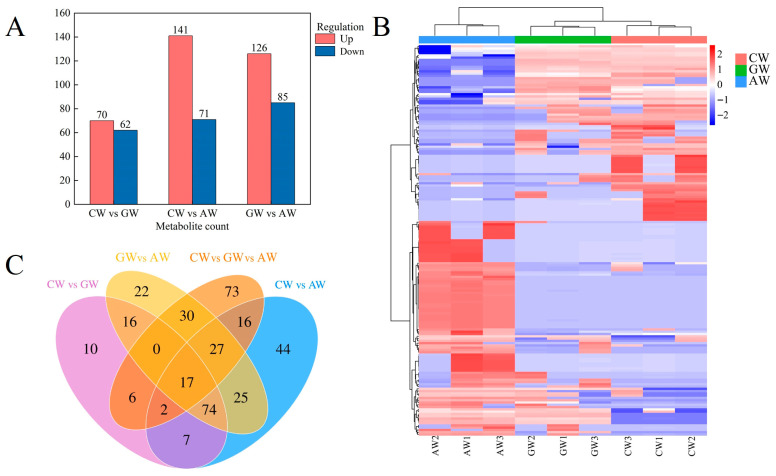
Terpene biosynthesis pathways. Expression levels of DEGs are reported in red–green scaled boxes, while accumulation of DAMs is reported in blue–red scaled boxes. For each transcript/metabolite, the abundance level is represented by a heat map, The bars from left to right represent three distinct types of fermented beverages: AW1–3: apple mono-fermentation; GW1–3: grape mono-fermentation; CW1–3: apple and grape co-fermentation. (**A**) Differential flavor compounds across comparison groups; (**B**) Distinct expression patterns of compounds among the three fermented beverage types; (**C**) Venn diagram illustrating shared and unique compounds in different comparison groups.

**Figure 4 foods-14-04119-f004:**
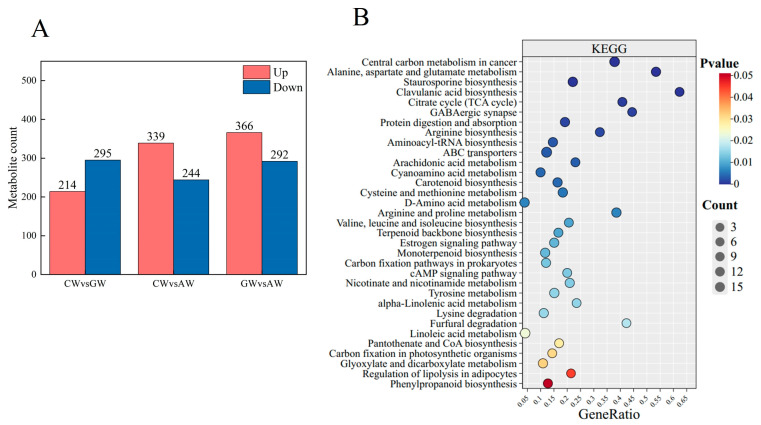
Differential metabolites in the three alcoholic beverages. (**A**) Number of differential metabolites in pairwise comparisons. (**B**) KEGG enrichment analysis of metabolites across wine types.

**Figure 5 foods-14-04119-f005:**
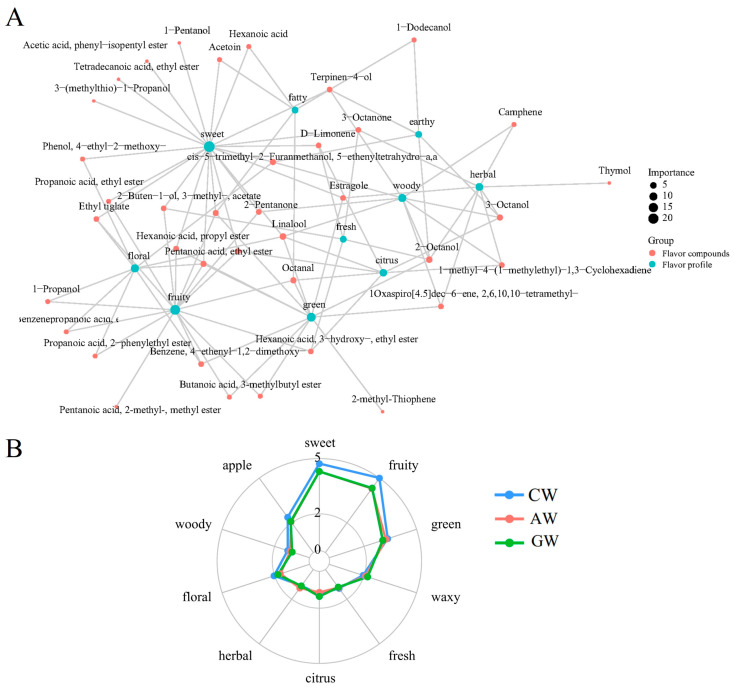
Network analysis of differential VOCs and associated sensory attributes. (**A**) Correlation network depicting interactions among CW, AW, and GW groups. Larger blue nodes represent sensory attributes connected to more VOCs; larger red nodes indicate VOCs contributing to multiple sensory attributes. (**B**) Sensory profiles derived from VOCs identified in the three wine types.

**Figure 6 foods-14-04119-f006:**
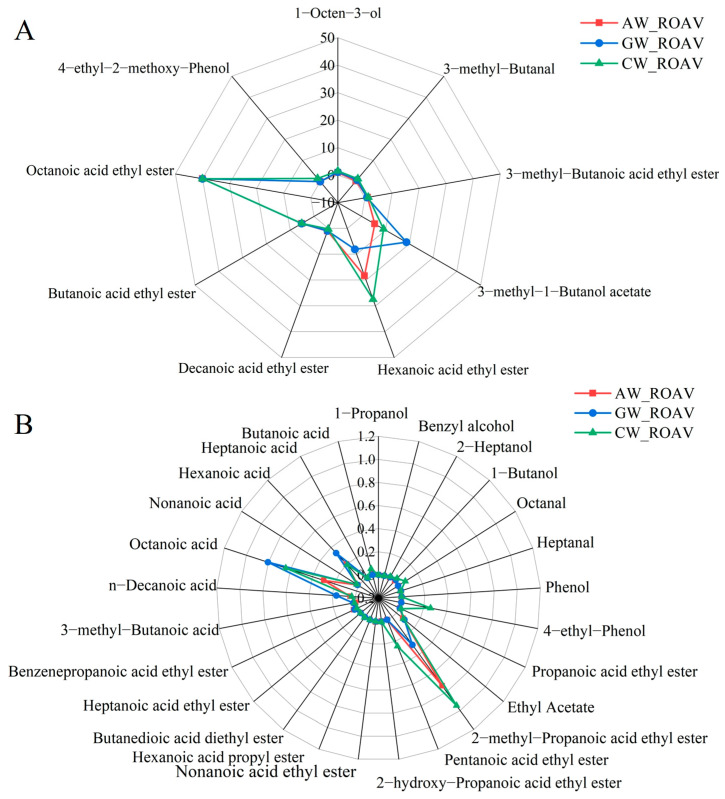
rOAV radar spectra of aroma compounds in three types of alcoholic beverages. (**A**) Representing compound rOAV > 1; (**B**) representing compound 0.1 < rOAV < 1.

**Figure 7 foods-14-04119-f007:**
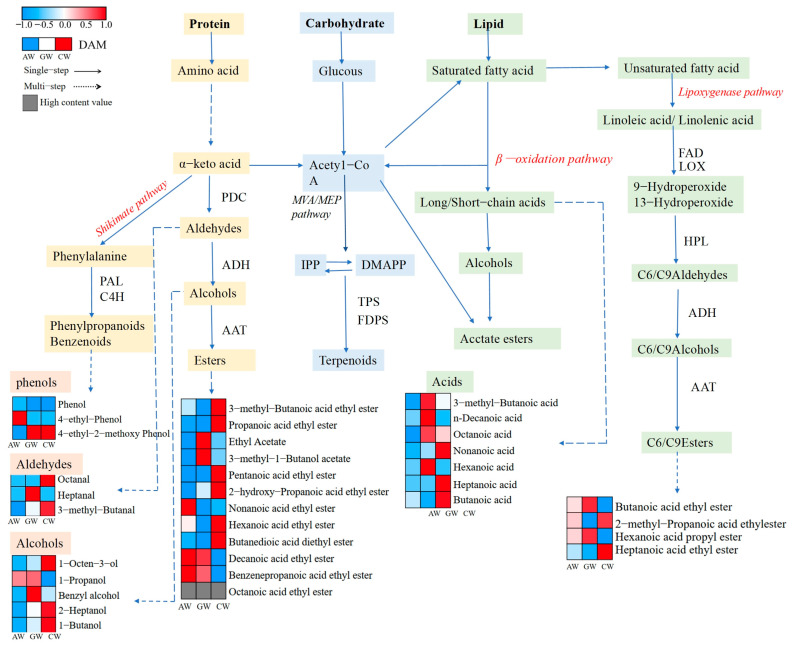
Major metabolic pathways of aroma-enhancing compounds (rOAV). Abbreviations: PDC, pyruvate decarboxylase; ADH, alcohol dehydrogenase; AAT, alcohol acyltransferase; FAD, fatty acid dehydrogenase; TPS, terpene synthase; LOX, lipoxygenase; HPL, hydroperoxide lyase; DMAPP, dimethylallyl pyrophosphate; IPP, isopentenyl pyrophosphate; PAL, phenylalanine ammonia-lyase; C4H, cinnamate 4-hydroxylase.

**Figure 8 foods-14-04119-f008:**
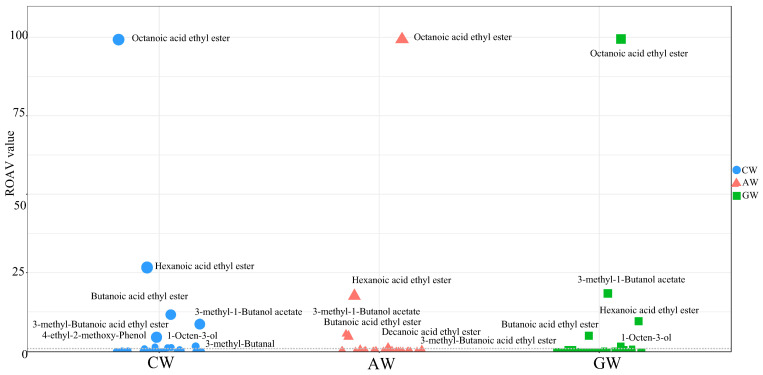
The ROAV Value of Different Class (dashed line: rOAV < 1).

**Figure 9 foods-14-04119-f009:**
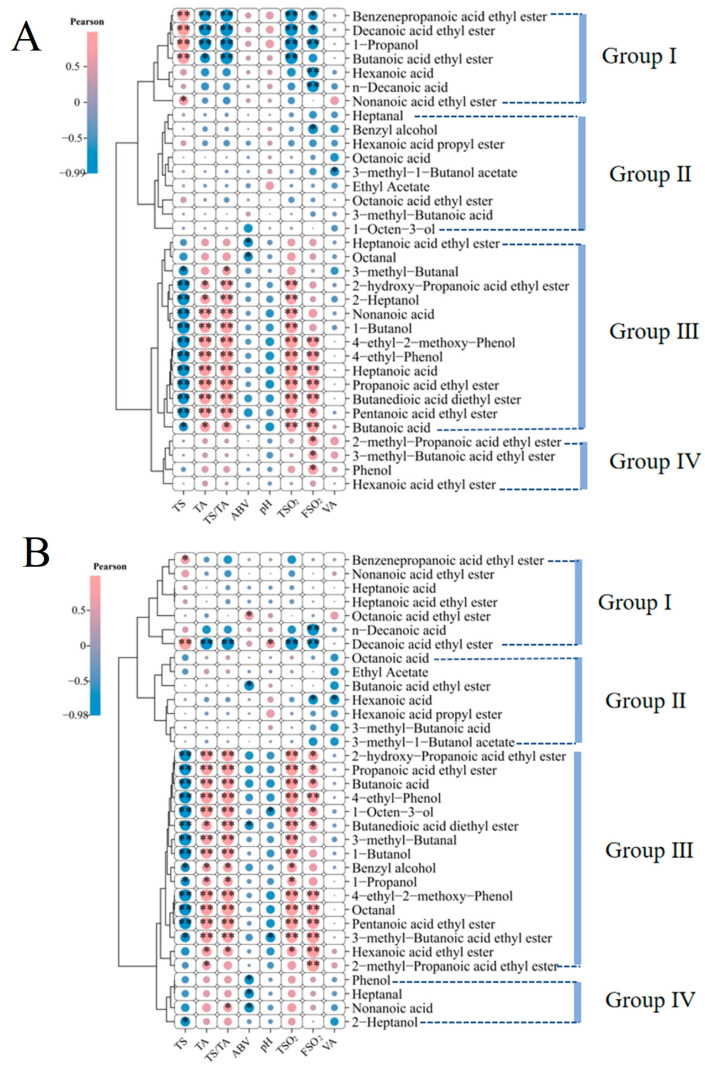
Correlations between aroma compounds and physicochemical parameters. (**A**) Correlation analysis of compound concentrations with physicochemical indices; (**B**) correlation analysis of rOAV values with physicochemical indices. ** *p* < 0.01, * *p* < 0.05. TS: total sugar; TA: titratable acidity; TS/TA: titratable acidity to total reducing sugars; ABV: alcohol by volume; TSO_2_: total sulfur dioxide; FSO_2_: free sulfur dioxide; VA: volatile acidity.

**Figure 10 foods-14-04119-f010:**
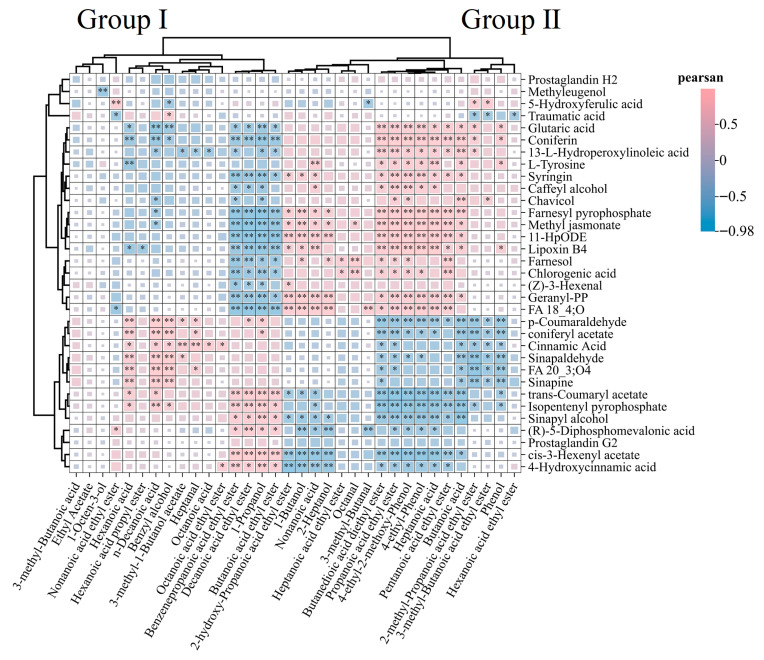
Correlation between VOCs and metabolites. ** indicates a significant level of *p* < 0.01; * indicates a significant level of *p* < 0.05. Horizontal axis: VOCs. Vertical axis: metabolites.

**Table 1 foods-14-04119-t001:** Physical and chemical properties of fruit samples.

Variety	Soluble Solids (°Brix)	Reducing Sugars (g/L)	Total Acidity (g/L)	pH
Italian Riesling	25.11 ± 1.09	231.33 ± 9.82	6.09 ± 0.07	3.65 ± 0.21
Sir Prize Apple	14.31 ± 0.32	116.52 ± 0.51	7.78 ± 0.95	3.43 ± 0.02
Grape juice–apple puree	22.82 ± 0.21	218.01 ± 1.22	5.61 ± 0.14	3.62 ± 0.07

Note: The total acidity of grape juice and grape–apple mixture is expressed as tartaric acid, while the total acidity of apple juice is expressed as malic acid. *n* = 3 (The independent biological experiment was repeated three times). Total acidity is calculated based on tartaric acid.

## Data Availability

The original contributions presented in this study are included in the article/Appendix A. Further inquiries can be directed to the corresponding authors.

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
