# Peer review of "Multi-Omics Elucidation of Flavor Characteristics in Compound Fermented Beverages Based on Flavoromics and Metabolomics"

_foods, 2025, doi:10.3390/foods14234119_

Round 1
Reviewer 1 Report
Comments and Suggestions for Authors
The current study comprises research focused on characterizing fruit wine compared to traditional single-fruit wine. The authors used analytical methods such as GC×GC–TOF MS and LC-MMS/MS to characterize the key odorants and metabolites of the product.
Regarding the research hypothesis, the study is novel and contributes to fruit wine research. Figures and tables are informative and acceptable.
Regarding the workflow of the paper, there are several problems. In particular, there are technical errors, including also and grammar issues.
-Abstract, line 21. Change ''offered'' to ''added''.
-Introduction, line 48. Do you refer to benzene-based compounds? The term aromatic in organic chemistry refers to benzene-based compounds. The term volatile is enough.
Line 52. Saccharomyces cerevisiae needs italic.
Line 83. Northwest region of China
Sections 2.2, 2.3, 2.7, and elsewhere. Revise lettering. There is different lettering.
Lines 205, 245, and elsewhere. Change ''detected '' to ''identified''.
Results and Discussion
Section 3.7. There is no clear connection with the information the authors want to give. Kindly revise it.
The same problem occurs in line 443.
Line 459. '' were significantly more abundant....''.
Authors contribution, funding, and conflict of interest statements are missing.
Finally, the authors must more explicitly provide the novely of their work compared to other similar studies.
Based on the above,
a major revision is suggested.
Comments on the Quality of English Language
The English language needs improvement.
Author Response
We extend our sincere gratitude for your valuable comments and suggestions, which have been immensely helpful in improving our manuscript. We have carefully addressed all the points raised in a point-by-point response. Should any further issues be identified, we warmly welcome your additional feedback. Please find the detailed responses in the attached document.

Reviewer 2 Report
Comments and Suggestions for Authors
I was tasked to review the draft manuscript "Multi-Omics Elucidation of Flavor Characteristics in Compound Fruit Wines Based on Flavoromics and Metabolomics" for the journal Foods. The manuscript is well written, clear, complete and supported by excellent graphs. I have only few comments:
- Line 41. The citation format should be [2,3]
- Line 73. The definition "unbiased" identification is too optmimistic. I rather prefer to define it "automatic" or something similar, or better to change the sentence keeping the meaning.
- Line 77. What is "integrated sistems biology approach"?
- Section 2.3. Check formatting style.
- Line 192. In addition to dynamic exclusion, it was important to implement an exclusion list based on blanks. Was it implemented?
- Section 2.6. Was the acquisition mode DDA?
- Section 2.7. Check formatting style.
These are just minor comments to be intended as minor revision request.
Author Response

(The authors gave the same response as above.)

Reviewer 3 Report
Comments and Suggestions for Authors
Line 23: Typo
Line 52: Saccharomyces cerevisiae needs to be italicized
Table 1: what is your sample size, what do the numbers mean, what acid was used to determine the total acid.
Line 105: While tartaric acid is the appropriate acid for grapes. Malic acid should be used to calculate TA for apples.
Section 2.2: Formatting. This section does not need to be italicized.
Section 2.3: Formatting. Font size
Section 2.3: What type of yeast did you use to make your wine? Did you propagate your yeast prior to pitching? What was your pitch rate? How many replications did you have for your fermentations? How big were your fermentations.
Section 2.7: Formatting. Does not need to be italicized.
Be consistent on your naming convention. You jump around from ethyl octanoate to octanoic acid, ethyl ester.
Line 431: I would not use the word co-fermentation to discuss the blending of multiple fruits into a wine. Co-fermentation is traditionally used to refer to the use of multiple microorganisms within the fermentation.
Author Response

(The authors gave the same response as above.)

Reviewer 4 Report
Comments and Suggestions for Authors
The manuscript presents an integrative flavoromics–metabolomics study investigating the compositional and sensory foundations of a compound apple–grape beverage in comparison with single-fruit beverage. The topic is relevant to Foods readers, addressing a growing niche in multi-fruit fermentation and demonstrating the potential of multi-omics approaches for product differentiation. The dataset is extensive, combining GC×GC-TOF-MS and LC-MS/MS analyses with statistical modeling and correlation mapping.
However, the work is somewhat descriptive rather than mechanistic, and several aspects require clarification or improvement.
Title and Introduction
- Please avoid using “wine” to refer to the mixed fermented beverage throughout; adopt a neutral term (e.g., “compound fermented beverage”) except when referring strictly to grape wine. This applies to the Abstract, Introduction, and Results text.
- The paper states novelty in combining flavoromics and metabolomics for compound fruit “wines.” Similar approaches exist for fruit blends/mixed fermentations (e.g., Liu et al., RSC Adv., 2022; Cao et al., Front. Nutr., 2022). Please specify precisely what is mechanistically or analytically new here (integration strategy, modeling, validation, or process design).
Materials and Methods
- Provide the physicochemical characteristics of the mixed initial must (grape juice + apple slurry, 4:1) to contextualize fermentation outcomes and TA/TS later discussed.
- Clarify whether independent biological fermentations were performed in triplicate (separate batches), or whether results are based on a single batch with analytical triplicates only; this materially affects statistical validity.
- Specify yeast strain, inoculation rate, nutrient additions, and fermentation kinetics (time-course, temperature control, and residual sugar at arrest). These details are essential for replication.
- Section 2.3 “aging”: The described steps (“filtered, aged, clarified, sterilized”) read as fining/clarification rather than true aging. Please rename or separate fining/clarification from any actual aging (conditions, duration, vessel).
- Data are reported as relative peak areas normalized to an internal standard, but no method validation is shown. At minimum, report linearity range and RSD% for key analytes; if available, add LOD/LOQ.
- You accept VOCs at match ≥ 700, RI ± 20, S/N ≥ 50. Indicate how many compounds were confirmed with authentic standards vs tentative identifications.
- You state data were normalized to total peak area or internal standard. Using total area can bias semi-quantitative across-matrix comparisons; please clarify that internal-standard normalization was consistently applied in both GC and LC datasets (and where it wasn’t, justify).
- “Adjusted p-values” are cited; specify the correction method (e.g., Benjamini–Hochberg FDR or Bonferroni) and where it was applied (OPLS-DA VIP filters, metabolite contrasts, etc.).
- Given the rOAV framework, include (or acknowledge as a limitation) a sensory panel and/or recombination/omission testing to validate aroma inferences. (The Abstract mentions recombination, but no data are shown.)
Results and Discussion
- Figure 1: The significant differences in residual sugar could largely stem from stopping fermentations at different endpoints (3–6 °Brix in Methods), which inflates between-sample diversity and complicates comparisons. Please justify endpoint selection and discuss this limitation.
- Figure 2: Define the compound classes grouped as “Others” and quantify their proportion. Given that “Others” appears to drive separation and the Venn distribution of specific compounds, transparency here is important.
- Current use largely shows “distinct groups,” which is expected from dissimilar substrates. Please provide a deeper interpretationwhich metabolic nodes/compound families load most strongly and drive separation? (Link to KEGG enrichments where relevant.)
- Very few compounds seem impactful (rOAV > 1). Consider splitting the radar into two panels (e.g., rOAV > 1 and 0.1 < rOAV ≤ 1) so lower contributors are visually interpretable.
- Cite the odor-threshold sources used to compute OAV/rOAV (e.g., specific databases or handbooks) and align them to Table S5 entries.
- The cross-correlation network (Fig. 10) is interesting but currently lacks quantitative metrics. Please report correlation coefficients and p-values, and provide a table of the top ~20 significant links to support the narrative. (Also cross-reference the supplementary network.)
- In the Discussion, soften deterministic phrases (e.g., “inevitable result,” “clear evidence”) and anchor claims directly to the presented statistics/plots.
Author Response

(The authors gave the same response as above.)

Reviewer 5 Report
Comments and Suggestions for Authors
Thank you for the opportunity to review the manuscript titled “Multi-Omics Elucidation of Flavor Characteristics in Compound Fruit Wines Based on Flavoromics and Metabolomics” by XiaoLong Li and co-authors. The study is methodologically advanced integrating flavoromics and metabolomics to elucidate flavor formation mechanisms in compound (apple–grape) fruit wine, providing analytical and biochemical insights into aroma formation. However, it needs improvement in certain aspects. Here are some specific comments:
Abstract
The addition of key numerical findings would strengthen the abstract.
Objectives
The aims of the study are stated. The authors should justify the selection of the specific varieties of grapes and apples, beyond availability.
Novelty
The study connects flavoromics with metabolic pathways, which is an emerging area. However similar multi-omics approaches have been used in fruit wine or cider research; thus, novelty lies mainly in the specific fruit combination and regional application.
Tables and figures
Table 1. Please add the organic acid used to express total acidity mentioned for grape and apple juice.
Figure 1. Please specify number of replicates, which test was performed to assess significant differences, and add what the abbreviations GW, AW and CW correspond to. The term total sugars should be changed to “reducing sugars”.
Methodology
2.2 Wine Quality Analysis :
- Please specify that certain methods were used for both juice and wine analysis.
- Could you provide references for the methods mentioned ?
2.3 Brewing Methodology
The authors should provide additional information regarding the equipment used (tanks – type, volume, number of replications for each sample), pneumatic pressure (type/ manufacturer used), and the fermentation parameters (yeast strain/manufacturer, nutrient supplementation-if any, manufacturers of enzymes and PVPP, duration of fermentation, oxygen control). The yeast strain, in particular, should be acknowledged for its critical role in shaping the aroma profile of the wines.
Also, handling of wine during ageing/ measures to protect from oxygen, duration of ageing, method of clarification, conditions of sterilization and equipment, as all affect the aroma of the produced wines.
Please specify how long after completion of fermentation were the samples analyzed and the storage conditions (temperature, atmosphere) until then. Were the samples bottled? In that case, please provide details. In line 174 is mentioned “thawing” of samples. Please elaborate.
Statistical analysis
What normalization method was applied to make the LC-MS and GC-MS results compatible for integrated analysis? The methods only state that each dataset was normalized to total peak area within itself.
Discussion
The discussion should include a more critical comparison with previous multi-omics wine studies referenced in this manuscript (e.g., references 11–17) or other relevant works.
Limitations
The authors should acknowledge the limitations of the study, eg. lack of sensory analysis.
Language
Minor grammatical errors should be corrected.
Author Response

(The authors gave the same response as above.)

Round 2
Reviewer 1 Report
Comments and Suggestions for Authors
The authors have now revised their paper and addressed my suggestions. Acceptance of the paper is now suggested.
Author Response
I sincerely thank you for your support and guidance during the review process. Your expertise and unique insights have been immensely beneficial, not only enhancing the quality of our work but also providing me with invaluable experience for my professional growth. Wishing you good health and success in all your endeavors.
Reviewer 3 Report
Comments and Suggestions for Authors
I feel the improvements to the manuscript are sufficient enough that it should be accepted at this time.
Reviewer 4 Report
Comments and Suggestions for Authors
I appreciate the careful revision of the manuscript from the authors, the overall clarity and organization have improved compared with the previous version. Nonetheless, there are still several methodological and reporting issues that should be addressed to ensure full transparency and internal consistency.
In Table 1, please present the grape juice and apple purée parameters with the same number of decimal places as those reported for the corresponding wine samples. Harmonizing the numerical precision across all entries will make comparisons more straightforward.
For total acidity, it would be helpful to report the values in a way that allows direct comparison between matrices. You may keep the native unit commonly used in your laboratory or local regulations, but please also provide a converted value in a standard unit (e.g., g/L as tartaric acid or malic acid equivalent) so that the acidity of grape juice, apple purée, and the resulting wines can be compared directly.
In the description of the yeast, the information about the strain and brand appears fragmented and partially repeated. At first mention, the brand information is omitted, but it is given later in the text. Please rewrite this paragraph to provide a concise, single description of the yeast (species, commercial name, manufacturer, and origin) and remove the redundant repetitions.
Please clarify whether any nutrient supplementation or other additives (e.g., yeast nutrients, vitamins, diammonium phosphate, tannins, or other oenological aids) were used during fermentation. If supplementation was applied, specify the type, dose, and stage of addition; if none was used, state this explicitly.
Fermentation temperature is currently described inconsistently, with both “15-18 °C” and “18 ± 0.5 °C” being reported. Please correct this discrepancy and state a single, consistent temperature regime (including whether there was an initial temperature range followed by strict control, or whether the fermentation was conducted entirely at 18 ± 0.5 °C).
The terminology used for sugar/solids-related parameters is confusing. In the caption of Figure 1, TS is defined as “reducing sugars”, whereas in Table 1 you list “Soluble solids” and “Reducing sugars” as separate variables, and in the text TS is sometimes described as “total soluble solids (TS)” and elsewhere as “total sugar”. Please standardize the nomenclature across the entire manuscript
Author Response
Thank you very much for your suggestion. We have responded to your suggestions one by one, as detailed in the attachment.

Reviewer 5 Report
Comments and Suggestions for Authors
The authors addressed most of my previous comments and the manuscript has been improved. However, the following points stilll need clarification of additional information.
Lines 125-128. Please provide some references.
Table 1. n=3, Please explain for the reader what n represents.
Lines 131-137. For Chinese standard methods please provide references.
Brewing Methodology: The content of this paragraph should be rearranged to improve flow and readability, especially the part with the manufacturers of products and equipment.
Lines 169-170: The authors mention that "Fermentations were carried out in 10 L stainless steel fermenters, each containing 8 L of grape must and 2 L of apple juice". However, in a 10 L fermenter, a working volume of 7.5 L to 8.5 L is usually recommended for safety and best practice. Please, clarify.
Line 426. β-Damascus ketones should be corrected to β-Damascenone
Author Response

(The authors gave the same response as above.)
